# A long-term assessment of the multidisciplinary degree of multidisciplinary journals

**Daniel Redondo-Gómez**[1]*, **Wenceslao Arroyo-Machado**[2], **Daniel Torres-Salinas**[2], **Antoni Margalida**[3,4], **Marcos Moleón**[1]

**1** Department of Zoology, University of Granada, Granada, Spain, **2** Department of Information and Communication Sciences, University of Granada, Granada, Spain, **3** Institute for Game and Wildlife Research, IREC (CSIC-UCLM), Ciudad Real, Spain, **4** Pyrenean Institute of Ecology (CSIC), Jaca, Spain

* drg@ugr.es

**Data Availability Statement:** Data can be found in https://doi.org/10.5281/zenodo.13269363.

**Funding:** DRG was funded by a predoctoral grant from the Junta de Andalucía (PREDOC_00262).

## Abstract

Are multidisciplinary journals truly multidisciplinary, and, how has the multidisciplinary character of these journals evolved over the long term? Here, we assess these questions by analyzing data from the Journal Citation Reports. We examined 983,246 articles and reviews published between 1980 and 2021 in 127 journals categorized under 'Multidisciplinary Sciences'. We found that the representation of the main branches of knowledge in multidisciplinary journals was uneven and, in general, not proportional to the global research effort dedicated to each branch. Similarly, the distribution of publications across specific research areas was uneven, with "Biochemistry & Molecular Biology" strongly overrepresented. However, we detected a decreasing trend in the percentage of publications that multidisciplinary journals dedicate to this and other top areas, especially over the last decade. The multidisciplinary degree of multidisciplinary journals, as measured by the Gini index, was generally low but showed a gradual increase over time. The impact factor of multidisciplinary journals was positively related to the percentage of publications in the area "Biochemistry & Molecular Biology". Compared to other multidisciplinary journals, *Nature*, *Science*, and *PNAS* emphasized this area even more strongly, though the difference between the first-ranked area and the other top areas consistently decreased since 1980. In conclusion, while a strong bias remains in favor of highly citable areas, multidisciplinary journals are progressively increasing their degree of multidisciplinarity in recent years. Thus, we encourage authors to carefully consider this polarization when selecting journals for their studies, and we suggest that scientific agencies keep it in mind when evaluating researchers.

## Introduction

Multidisciplinary journals, i.e., those listed in the Journal Citation Reports (JCR) under the category 'Multidisciplinary Sciences' or in Scopus within the subject area 'Multidisciplinary', are

WAM was funded by FPU Grant (FPU18/05835) from the Spanish Ministry of Universities. MM was partly supported by the Severo Ochoa Program for Centres of Excellence in R+D+I (SEV-2012-0262) and by a research contract Ramón y Cajal from the MINECO (Ministerio de Economía, Industria y Competitividad; RYC-2015-19231). The funders had no role in study design, data collection and analysis, decision to publish, or preparation of the manuscript.

**Competing interests:** The authors have declared that no competing interests exist.

supposed to accommodate the publication of research papers from various areas of research or disciplines. But, to what extent are these journals actually multidisciplinary? In these journals, multidisciplinarity may be achieved through two ways: by including multidisciplinary publications and by gathering unidisciplinary publications from multiple disciplines. However, multidisciplinary journals often publish a small proportion of truly multidisciplinary articles [1] and are greatly biased towards certain disciplines [2, 3]. In particular, multidisciplinary journals usually tend to accept articles based on their potential citability [4, 5]. Thus, the multidisciplinarity of these journals may be compromised by fostering a few areas characterized by high-impact scientific production [3] to the detriment of less cited areas.

Though multidisciplinary journals have been profusely analyzed in the scientific bibliometric literature, most research has been focused on very specific issues or only on a few top journals [2, 3, 6]. The increase in the multidisciplinarity of several specific areas is well documented (such as medicine [7]; cognitive science [8]; nanoscience and nanotechnology [9]; and others [10]) but little is known about how the multidisciplinary character of multidisciplinary journals has changed over time. Despite this trend in specific areas, the growing competitiveness of the increasingly crowded multidisciplinary category may jeopardize its multidisciplinary nature, since it is expected that journals use all possible resources to become and remain as high as possible in the impact and citation rankings [4].

Here, we aim to explore how the multidisciplinary character of multidisciplinary journals has changed in the long term. For this general purpose, we have delineated five main specific goals at three levels. First, at the level of the *'Multidisciplinary Sciences'* category, we aim i) to determine if the representation of the different branches and areas of knowledge in multidisciplinary journals is proportional to the null expectations indicating multidisciplinarity. Our null expectations are: a) multidisciplinary journals allocate the same space (i.e. similar proportion of items published) to each branch/area, and b) the space allocated in multidisciplinary journals to each branch/area is proportional to the global research effort behind each branch/area. Also, we aim ii) to identify the areas of knowledge that have been overrepresented in multidisciplinary journals, and if they have changed over the last four decades. Second, at the level of all *multidisciplinary journals*, we aim iii) to assess which journals are more multidisciplinary and if they have changed over time. Also, we aim iv) to investigate if a) the multidisciplinarity degree of the journal and b) the proportion of publications in the journal that belong to the top area (according to the number of articles) are related to the impact factor of multidisciplinary journals. Third, with a focus on the three *top multidisciplinary journals* (*Nature*, *Science*, and *Proceedings of the National Academy of Sciences*, *PNAS*), we aim v) to establish which of them is advancing the most toward multidisciplinarity.

## Materials and methods

### Web of Science categories

Web of Science and InCites have different classification schemas for the research categories of scientific journals, with the categories of Web of Science being the most used. This scheme, comprising a total of 254 categories, is used to classify the journals indexed in the Journal Citation Reports (JCR). These journals can be assigned up to six categories based on criteria such as thematic alignment, author and institution affiliations, citations and references (https://support.clarivate.com/ScientificandAcademicResearch/s/article/Web-of-Science-Core-Collection-Web-of-Science-Categories?language=en_US). Additionally, these categories are then inherited by the articles published in them. However, there is an exception for the categories "Multidisciplinary Sciences" and "Medicine, General, and Internal". Their articles are reassigned through an algorithmic process based on their references to other thematic areas,

although this is only done in InCites and not for all publications (https://incites.help.clarivate.com/Content/Indicators-Handbook/ih-document-reclassification.htm).

## Data gathering

First, we selected all journals included in the category "Multidisciplinary Sciences" for at least one of the editions between 1997 and 2020 (n = 127 journals; "multidisciplinary journals"), according to data from the Journal Citation Reports. We excluded ESCI (Emerging Sources Citation Index) journals. The journals were individualized by ISSN/eISSN, assigning the most recent name in case of name changes. Of these journals, 55 (43%) were included in the category "Multidisciplinary Sciences" for at least half of the study period, and only 10 (8%) were included for <5 years. Also, during the study period, 86 journals (68%) were exclusively categorized as "Multidisciplinary Sciences". Then, we gathered all publications, i.e., "articles" and "reviews" (documents that always undergo peer review and receive the majority of the citations), from these journals between 1980 and 2021, after manually filtering the journals by ISSN in the InCites publication sources module and downloading all publications that met the criteria for time and typology. From these publications, we recorded the publication date (year), source (publishing journal), and research "area" (Web of Science categories; note that InCites algorithmically reassigns publications in multidisciplinary journals to their most relevant area, accordind to their cited references when available). Following Arroyo-Machado & Torres-Salinas, 2021 [11], we classified the current 254 research areas into six major "branches" of knowledge: "Multidisciplinary", "Arts & Humanities", "Life Sciences & Biomedicine", "Physical Sciences", "Social Sciences", and "Technology". It is important to note that the "Multidisciplinary" branch includes only one area ("Multidisciplinary Sciences"). Second, we gathered all publications from all JCR journals for the same period (1980–2021). From these publications, we also recorded the publication date (year), source (publishing journal), "area", and "branch".

## Data treatment and analyses: Multidisciplinary sciences

For each research area, we calculated 1) the number of publications in the research area in all multidisciplinary journals each year, as a measure of the gross number of publications, 2) the ratio between the percentage of publications in each research area each year in multidisciplinary journals and the percentage of publications in each research area each year in the world (hereafter, "multidisciplinary/world ratio"), as a measure of the weighted number of publications relative to the global research effort and 3) the position of the research area in the ranking of the number of publications by multidisciplinary journals each year. Publications included in more than one research area were considered in each of the areas they were included (but note that only 3.1% of publications were included in more than one area, and that these multiple areas were often closely related, e.g., "Computer Science, Hardware & Architecture" and "Computer Science, Theory & Methods"). To test our null expectations indicating multidisciplinarity, we performed Chi-squared analyses comparing, for the entire study period and excluding the branch "Multidisciplinary Sciences", the median percentage of articles published in each branch in multidisciplinary journals and a) a balanced percentage (i.e., 20% of total publications belonging to each of the five branches) and b) the percentage of articles published in each branch in the world (i.e., in all journals except those included in the area "Multidisciplinary Sciences"). To explore inter-annual trends, we made Pearson correlations between the year and the yearly percentage of publications that multidisciplinary journals devote to a) the top-30 areas of knowledge (according to the total number of publications) and b) the area "Biochemistry & Molecular Biology".

### Data treatment and analyses: Multidisciplinary journals

Similarly, for all multidisciplinary journals (i.e., n = 127 journals within the area "Multidisciplinary Sciences"), we calculated 1) the number of publications in each research area each year, 2) the ratio between the number of publications in the journal and the world number of publications in each research area each year (hereafter journal/world ratio), and 3) the "Multidisciplinarity degree" (from the Gini index) [12, 13]. This inequality index ranges between 0 (all research areas are equally represented i.e., they have the same number of publications) and 1 (all publications are concentrated in one research area). The Gini index was computed using the function ineq() from the ineq package [14]. The multidisciplinarity degree was calculated as 1—Gini index. Additionally, we made Pearson correlations between the yearly impact factor of multidisciplinary journals (according to Web of Science data) and a) their yearly multidisciplinarity degree and b) the yearly percentage of publications belonging to the top-ranked research area: "Biochemistry & Molecular Biology". Also, we correlated the yearly percentage of publications belonging to "Biochemistry & Molecular Biology" with the yearly multidisciplinarity degree. To detect temporary trends, we made additional Pearson correlations between the yearly percentage of publications belonging to "Biochemistry & Molecular Biology", the percentage of publications in the top-30 areas (according to the number of articles and excluding "Multidisciplinary Sciences") and the year (as a proxy of time). All statistical analyses were conducted in R 4.0.5 [15].

### Data treatment and analyses: Top multidisciplinary journals

The world's three most influential multidisciplinary journals: *Nature*, *Science*, and *PNAS* were analyzed in detail. As for the rest of the journals, we calculated 1) the number of publications in each research area each year, 2) the journal/world ratio, and 3) the "Multidisciplinarity degree" from the Gini index, as well as the temporal changes of these variables throughout the study period. To detect temporary trends, we also made Pearson correlations between the yearly percentage of publications belonging to "Biochemistry & Molecular Biology", the percentage of publications in the top-30 areas (according to the number of articles and excluding "Multidisciplinary Sciences") and the year (as a proxy of time).

## Results

### Relative role of research branches and areas in the 'Multidisciplinary Sciences' category

Our bibliographic dump of the 'Multidisciplinary Sciences' category included 127 journals that published a total of 983,246 articles and reviews (hereafter "publications") between 1980 and 2021 (S1 Table). More than half of these publications (604,578 publications; 61.5% of the total) belonged to the branch of knowledge "Life Sciences & Biomedicine", followed by "Physical Sciences" (152,629 publications; 15.5%; S1 Table). The branch with the lowest representation was "Arts & Humanities" (6,124 publications; 0.6%; S1 Table). The distribution of publications by branches of knowledge in multidisciplinary journals was clearly uneven, either for the entire study period ($\chi^2$ = 40.51, p < 0.001 in the Chi-squared test comparing the median percentage of publications published in each branch and the balanced percentage, i.e., 20% for each of the five branches, excluding "Multidisciplinary Sciences") or separately by decades ($\chi^2$ = 37.78–47.32, p < 0.001 for all cases). In addition, we found differences between the median percentage of articles published in each branch in multidisciplinary journals with respect to the median percentage of articles published in each branch in the world, for the entire period ($\chi^2$ = 15.15, p < 0.01) and for 1980–1989 ($\chi^2$ = 14.89, p < 0.01), 2010–2019 ($\chi^2$ =

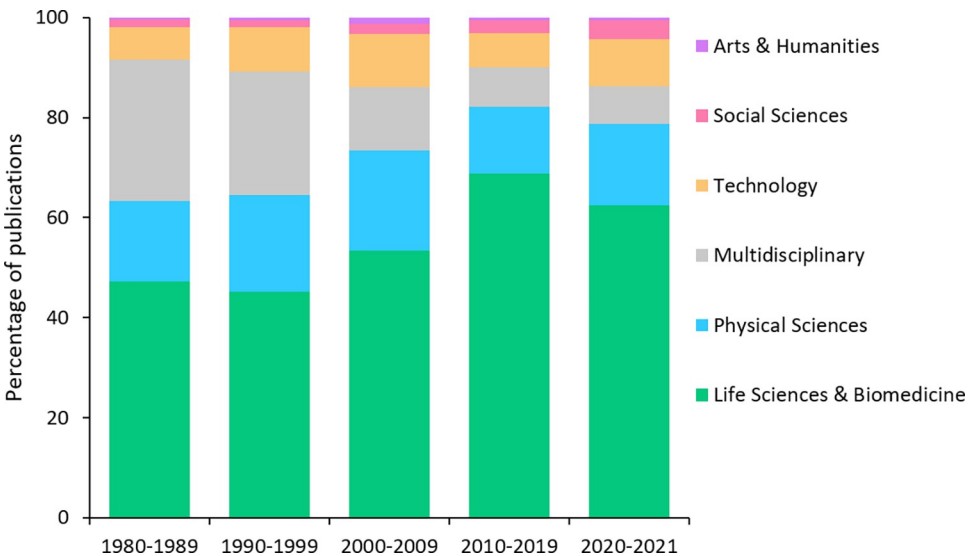

**Fig 1. Decadal evolution of the percentage of publications ("articles" and "reviews") per branch of knowledge in multidisciplinary journals (according to the Web of Science) from 1980 to 2021.**

23.23, p < 0.001) and 2020–2021 ($\chi^2$ = 14.70, p < 0.01), but not for 1990–1999 ($\chi^2$ = 8.24, p = 0.083) and 2000–2009 ($\chi^2$ = 6.85, p = 0.144). Since 1980, the relative contribution of the branch "Multidisciplinary" to multidisciplinary research has been decreasing in favor of "Life Sciences & Biomedicine", while the relative contribution of the rest of the branches of knowledge has remained approximately stable (Fig 1).

With regards to the research areas (Fig 2), the 30 areas with the highest number of publications belonged to the branches "Multidisciplinary Sciences" (note that there is only one area within this branch), "Life Sciences & Biomedicine" (43.3% of the 30 areas), "Physical Sciences" (40.0%), "Technology" (10.0%) and "Arts and Humanities" (3.3%; there was none research area belonging to the branch "Social Sciences" within these 30 areas). "Multidisciplinary Sciences" and "Biochemistry & Molecular Biology" were the two areas with more publications per year in multidisciplinary journals (12.0% and 13.1% as average of total publications in multidisciplinary journals, respectively; Fig 2B). These areas were followed by other areas belonging to the branches of knowledge "Life Sciences & Biomedicine" and "Physical Sciences", such as "Ecology", "Neurosciences", "Immunology", "Mathematics", "Geochemistry & Geophysics" and "Cell Biology". Within the branch "Technology", the research areas of "Engineering, Multidisciplinary" and "Materials Science, Multidisciplinary" were the two areas with more publications. Within the branch "Arts and Humanities", "History & Philosophy of Science" was the only research area among the 30 areas with more publications. Overall, we detected a strong imbalance among areas, as the number of publications relating to the first five top areas was similar to the total number of publications relating to all the remaining areas in the top 30 (median = 29,297 publications, corresponding to the fifth area "Immunology"). Moreover, the average yearly number of publications belonging to "Biochemistry & Molecular Biology" (3058 publications, range = 1401–7381, 13.1%) was markedly higher than the average yearly number of publications belonging to each of the other first 30 areas excluding "Multidisciplinary Sciences" (395 publications, range = 66–1543, 0.3%). In general, the percentage of publications that multidisciplinary journals devoted to these top-30 areas has increased since 1980 (r = 0.798, p < 0.001), although this trend was reversed since 2011 (S1 Fig). In the case of the area "Biochemistry & Molecular Biology", its percentage in multidisciplinary journals has been

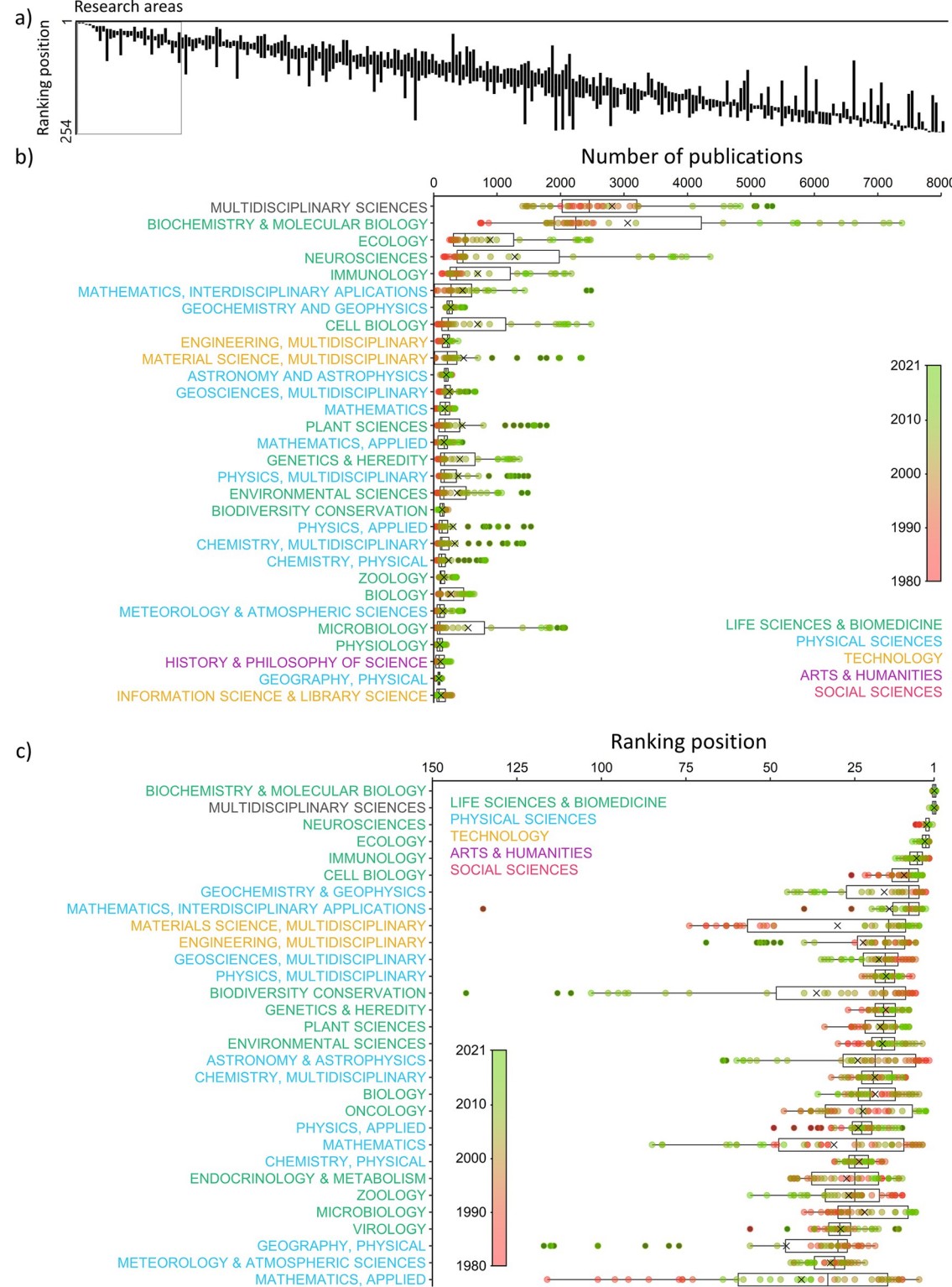

**Fig 2.** a) Position of each research area in the ranking of the number of publications in multidisciplinary journals per year. Number one in the y axis represents the research area with more publications. Graph a) represents the complete graph including all the research areas and the graph c) is the enlarged part indicated by the square (with the axes inverted). b) Number of publications of each research area in multidisciplinary journals per year. Only the first 30 research areas for each graph are shown, decreasingly sorted by the median. Research areas are classified by branches of knowledge and points are colored according to year.

decreasing since the mid-1980s (r = -0.548, p < 0.001; S1 Fig). This decrease has been particularly marked in the last decade (r = -0.975, p < 0.0001; S1 Fig).

When sorting the research areas by the number of publications per year, we found that the top-ranked areas for all years during the entire study period were "Biochemistry & Molecular Biology" and "Multidisciplinary Sciences" (Fig 2A and 2C), except in 2012, when "Biochemistry & Molecular Biology" and "Neurosciences" were in the top positions. Before 1999, "Multidisciplinary Sciences" was always in the first position followed by "Biochemistry & Molecular Biology", but, since 2000, these positions became reversed, except in 2021, in which "Multidisciplinary Sciences" returned to the first position. The following areas have suffered more inter-annual changes in their position in the ranking, especially as their median position in the total ranking is lower. Some areas showed a clear increasing trend in their positions in the ranking, such as "Neurosciences", "Cell Biology" and "Materials Science", while others showed the opposite trend, such as "Geochemistry & Geophysics", "Geosciences" and "Astronomy & Astrophysics" (Fig 2C).

The overwhelming primacy of "Multidisciplinary Sciences" and "Biochemistry & Molecular Biology" was also evident when plotting the percentage of publications of each area in multidisciplinary journals against the world percentage of publications of each area (Fig 3A and 3B). To a lesser extent, other areas such as "Ecology", "Neurosciences", "Immunology", "Mathematics" and "Cell Biology" were also overrepresented. In contrast, other research areas such as "Surgery", "Engineering, Electrical & Electronic", "Chemistry, Multidisciplinary", "Physics, Applied" and "Material Sciences" were underrepresented (see Fig 3A and 3B). "Virology", "Immunology", "Neurosciences" and "Cell Biology" consistently increased their representation over time with respect to the world context. Besides, other areas such as "Biodiversity Conservation", "Geography, Physical" or "Astronomy and Astrophysics" have lost prominence since 1980 (Fig 3C).

## How multidisciplinary are multidisciplinary journals, and its relation to the journal impact

First, we assessed which journals are more multidisciplinary by calculating their "Multidisciplinarity degree" from the Gini index. In general, the multidisciplinarity degree was low in all multidisciplinary journals during all years (<0.25 in all cases; Fig 4). Specifically, two thirds (67.7%) of multidisciplinary journals showed an average Gini index higher than 0.95. *Royal Society Open Science*, *PLoS ONE*, *Scientific Reports*, and *PEERJ* were the journals with the highest median multidisciplinarity degree (lowest Gini index). The general trend for the older journals was to increase their multidisciplinarity degree over time. *Nature*, *Science*, and *PNAS* appear in the ranking positions as 19th, 29th, and 44th, respectively (Fig 3).

The journals' impact factor was positively related to their percentage of publications in the area "Biochemistry & Molecular Biology" (r = 0.290, p < 0.001; Fig 4C), as well as to their multidisciplinarity degree (r = 0.200, p < 0.001; Fig 4D). Accordingly, there was also a positive correlation between the percentage of publications in the area "Biochemistry & Molecular Biology" and the multidisciplinarity degree of the journals (r = 0.334, p < 0.001; Fig 4E). However, this counterintuitive correlation is conditioned by a few values, and journals that published many "Biochemistry & Molecular Biology" publications showed in general very low multidisciplinarity degrees (Fig 4E). In fact, there were no journals with many "Biochemistry & Molecular Biology" publications and high multidisciplinarity degree.

## Multidisciplinarity in top multidisciplinary journals

As observed for the whole "Multidisciplinary Sciences" category, the number of publications of the different research areas in the top multidisciplinary journals was highly uneven (S2 Fig).

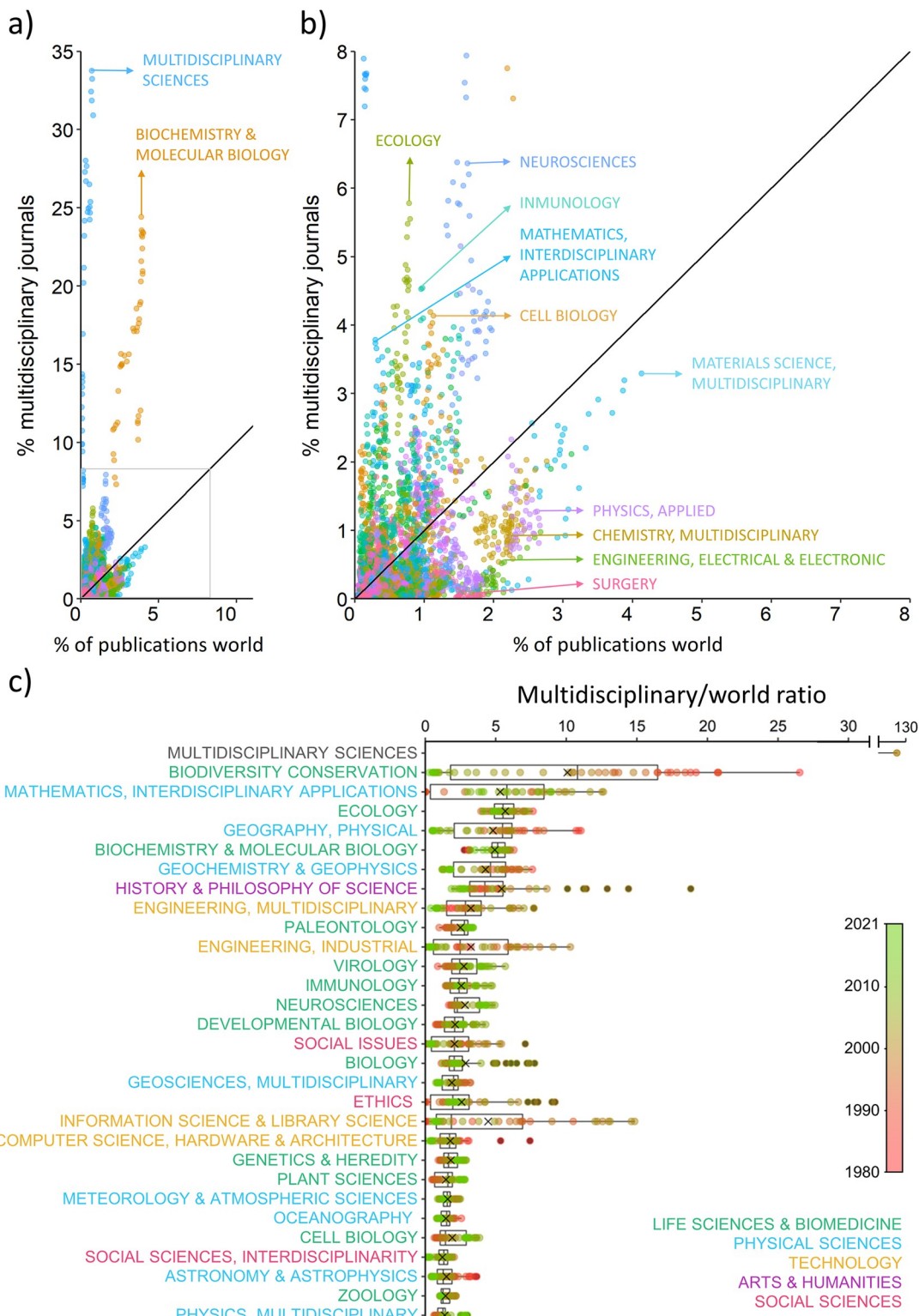

**Fig 3. Relation between the percentage of publications of each research area in multidisciplinary journals and the percentage of publications of that research area in the world.** Graph b) is a zoom of the indicated part of graph a). Each research area is represented by a color. The black line represents the 1:1 ratio, namely the publications in each research area in multidisciplinary journals are proportional to the global publications in that area. Note the great overrepresentation of " Biochemistry & Molecular Biology " in multidisciplinary journals. c) Ratio between the percentage of publications in each

research area each year in multidisciplinary journals and the percentage of publications of each research area each year in the world (multidisciplinary:world ratio).

However, unlike for the "Multidisciplinary Sciences" category, "Biochemistry & Molecular Biology" was the research area with the highest number of publications in the three top multidisciplinary journals, well above "Multidisciplinary Sciences", especially in *PNAS*. In particular, "Biochemistry & Molecular Biology" represented 27.2–47.7% of total publications of the top 15 areas in these journals. We found that the number of publications relating to the first two top areas was similar to the total number of articles relating to all the remaining areas in the top-15 (median = 3395 publications, corresponding to the third area "Astronomy and Astrophysics" for *Nature*; 2755 publications, corresponding to the third area "Neurosciences" for *Science*; and 6964 publications, corresponding to the third area "Multidisciplinary Sciences" for *PNAS*). These results were consistent between journals and over time (see S2 Fig). That said, the representation of "Biochemistry & Molecular Biology" in these three journals is consistently decreasing since 1980 (r = -0.649, p < 0.001 for *Nature*; r = -0.461, p < 0.01 for *Science*; r = -0.988, p < 0.001 for *PNAS*), though still these journals publish more than double of publications of this area compared to the average for other multidisciplinary journals. Inversely, the representation of the other top research areas is increasing in recent years, especially in *PNAS* (S2 Fig). Focusing on the 15 areas with the highest number of publications, most of the publications were within the branch of knowledge of "Life Sciences & Biomedicine" (88.6% for *PNAS*, 57.0% for *Nature* and 52.4% for *Science*), while *Nature* and *Science* also had a large number of publications in "Physical Sciences" (28.0% for *Nature*, 29.8% for *Science* and 4.0% for *PNAS*; S2 Fig). Comparing with all multidisciplinary journals, there are some areas that play a notably more (e.g., "Astronomy & Astrophysics") and less prominent role (e.g., "Ecology" and "Mathematics, Interdisciplinary Applications") in the three top journals.

With regards to the ranking of each area in the three top multidisciplinary journals, we observe a similar inter-annual pattern than for all multidisciplinary journals (Fig 2C), though some differences arise. For instance, in *Nature*, the position in the ranking of the top areas is much more static (S3 Fig). When weighing the number of publications by the world number of publications in each research area, the prominence of the branches of knowledge of "Life Sciences & Biomedicine" and "Physical Sciences" is maintained. In particular, "Biochemistry & Molecular Biology" and "Geochemistry & Geophysics" were research areas with a clear overrepresentation. However, there is a marked and ubiquitous tendency to reduce the overrepresentation of top research areas, especially in *Nature* and *Science*. In addition, *PNAS* shows a clear increase in the representation of underrepresented areas (Fig 2C and S4 Fig).

## Discussion

Here, we present a comprehensive long-term analysis of the multidisciplinary character of multidisciplinary journals. Excluding the "Multidisciplinary" branch and the corresponding "Multidisciplinary Sciences" area, which accounted for 12% of total publications in these journals, we observed a strong bias in favor of publications within specific branches and areas of knowledge, consistent with findings from previous studies [1–3]. Nonetheless, our analysis also reveals a recent trend in multidisciplinary journals toward reducing this imbalance.

Contrary to our null expectations, we found that the representation of the five main branches of knowledge, namely "Life Sciences & Biomedicine", "Physical Sciences", "Technology", "Social Sciences", and "Arts & Humanities" (in that order according to their number of publications), in multidisciplinary journals was not fully balanced nor, except for the period 1990–2009, proportional to the global research effort dedicated to each branch. Thus,

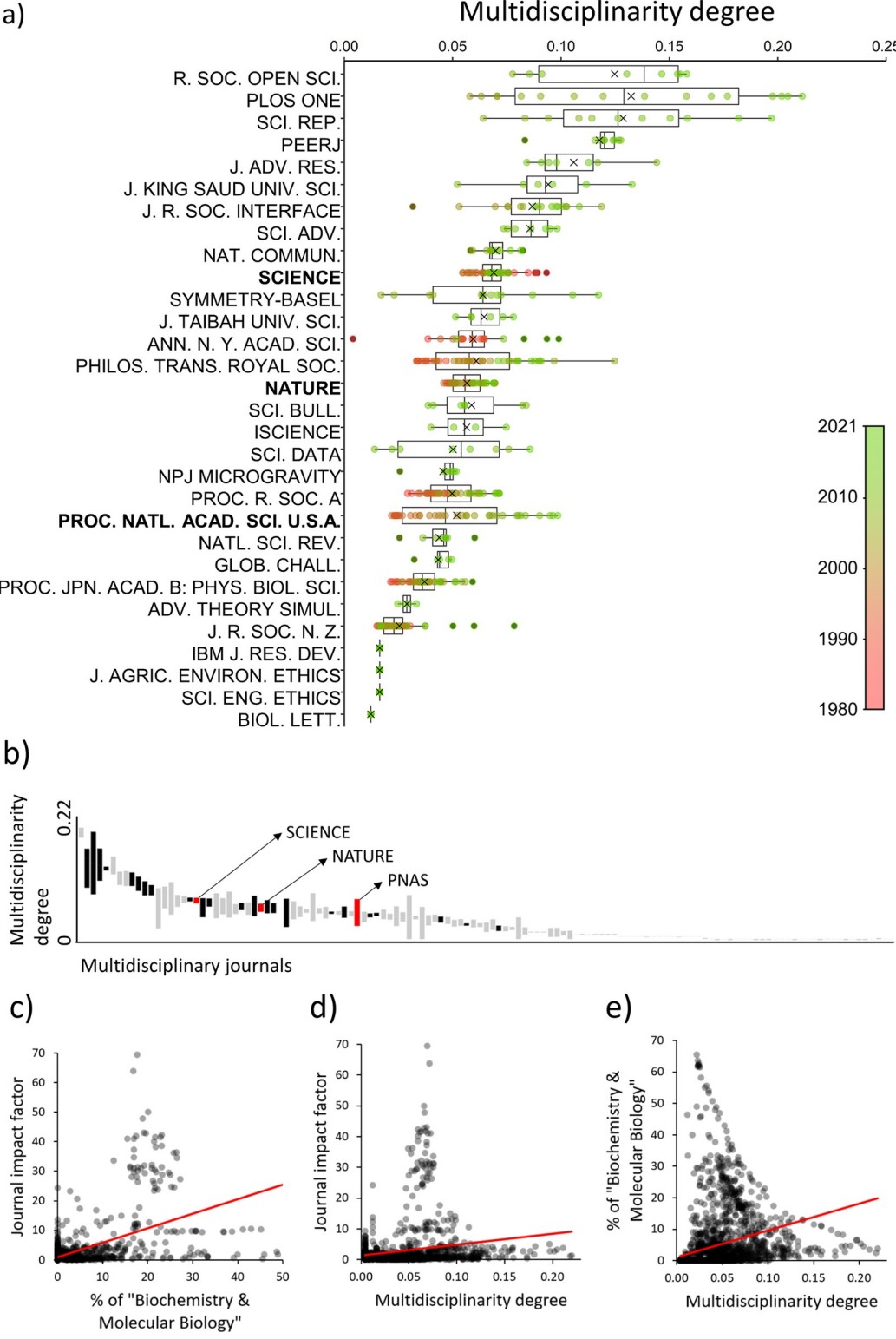

**Fig 4. Yearly multidisciplinarity degree in multidisciplinary journals and correlations.** In graph a), only the first 30 Q1 and Q2 multidisciplinary journals are shown, decreasingly sorted by the median. The journals in bold are top multidisciplinary journals and points are colored according to year. In graph b), all multidisciplinary journals are represented, decreasingly sorted by the median. Black bars represent Q1 and Q2 journals and grey bars represent Q3 and Q4 journals. Top multidisciplinary journals (*Nature*, *Science*, and *PNAS*) are represented by red bars. c) Relation between

the percentage of publications in the area "Biochemistry & Molecular Biology" in each multidisciplinary journal and the journals' impact factor. d) Relation between the multidisciplinarity degree of multidisciplinary journals and the journals' impact factor. e) Relation between the multidisciplinarity degree of multidisciplinary journals and the percentage of publications in the area "Biochemistry & Molecular Biology" in each multidisciplinary journal. The red lines represent the linear trend lines.

multidisciplinary journals may be regarded as multidisciplinary insofar their scientific production covers several branches of knowledge, but not because they allocate a balanced or proportional number of publications among branches [6]. In particular, publications in multidisciplinary journals were strongly biased toward research areas where state-of-the-art research and paradigm-busting work are common. Out of the 254 areas included in the Journal Citation Reports, the percentage of publications in multidisciplinary journals belonging to the top-30 areas (according to the number of publications and excluding "Multidisciplinary Sciences") increased from c. 42.3% in 1981 to c. 67.7% in 2012. However, this trend was decreasing since 2011, reaching c. 57.3% in 2021, which suggest a new tendency towards a more even distribution of publications among the different areas of knowledge. Within these 30 top-areas, there is a strong overrepresentation of the area "Biochemistry & Molecular Biology" (Figs 2 and 3). Multidisciplinary journals, especially *Nature*, *Science* and *PNAS*, are consistently decreasing the number of publications and overrepresentation of this area, although its prevalence is still very high, especially for the three top journals (S1, S2 and S4 Figs). Overall, for multidisciplinary journals, favoring this and other top areas means in general increasing also their impact factor–and success [4, 5]. In this line, we found a positive relationship between the impact factor of multidisciplinary journals and the percentage of their publications that is devoted to "Biochemistry & Molecular Biology".

The current system of publication and evaluation of journals often rewards the most polarized multidisciplinary journals, while journals seeking higher balance among disciplines are generally penalized in terms of impact factor (Fig 4) [16]. More equitable multidisciplinary journals usually receive fewer citations and have lower impact factor compared to journals that focus on high-impact areas (Fig 4) [17]. The scientific value of multidisciplinary research has been widely highlighted in the scientific literature [18] and many authors argue for its positive effects on the impact and citability of articles [19–23]. Although our results are not inconsistent with this idea, they reveal that the journals with the highest impact (journal impact factor >25) publish at least 15% of their articles in "Biochemistry & Molecular Biology" (Fig 4C) and multidisciplinarity has a lower effect on the impact of journals (Fig 4D). Thus, being multidisciplinary can have a positive effect on the impact of journals (Fig 4D) but publishing in leading areas is much more effective (Fig 4C). However, the temporal evolution of the degree of multidisciplinarity observed in some multidisciplinary journals demonstrates that more multidisciplinary and balanced publication distribution does not always imply a lower scientific impact. For instance, the degree of multidisciplinarity in *PNAS* has consistently increased (by reducing its publications in "Biochemistry & Molecular Biology") since 1980 while increasing its impact factor.

For authors, editors, and scientific entities, this polarization should be a matter of concern for several reasons. First, it can generate terminological confusion because many journals whose scientific production is highly biased towards a single area are under the label of "multidisciplinary" (see Fig 4A). Many science institutions and agencies utilize these categories for rankings, journal evaluations, and even personnel selection within the academic career. Thus, a mismatch between a journal's scope and its assigned category could result in inaccurate rankings and evaluations. While the multidisciplinary label need not be removed from these journals, a greater alignment between the actual scope of a journal and its assigned category would be beneficial. Whether this effort should be mostly made by Clarivate (e.g., by reconsidering

the categories and the assignment criteria), the journals (e.g., by broadening the range of disciplines represented in their publications and featuring predominant areas within specialized journals), and/or the agencies (e.g., by weighing rankings and evaluations according to the existing bias) is a matter for further discussion.

Second, some multidisciplinary articles that do not fit in specialized journals can have a significant barrier to being published since they have to compete with highly specialized articles that are much more attractive for the success of the journal [6]. This results in greater difficulty and time investment to publish multidisciplinary articles [24], not always rewarded with more impact or success [12, 25]. Thus, this polarization jeopardizes some of the important functions that multidisciplinary journals could fulfill, such as stimulating innovative ideas or providing cohesion among disciplines [6]. Multidisciplinary research is also handicapped because it has fewer chances to be funded than more narrowed research [26]. These issues raise some fundamental questions: Is it primarily the journal, the editor, or the reviewers who shape the journal's focus? Does the degree of multidisciplinarity shift with changes in editorial leadership? Are there intentional decisions to exclude certain articles, thereby limiting the publication of more multidisciplinary work? Addressing these questions is essential for understanding the factors that shape the multidisciplinary degree of these journals and could stimulate future debate and research on this topic.

Third, the recurrent focus on highly cited and populated fields such as those abovementioned, may be influenced by editorial strategies aimed at increasing a journal's JIF (Journal Impact Factor), which is often used as a proxy for academic relevance, rather than by a genuine interest in covering a broad range of disciplines. This can marginalize research that, although valuable, may generate fewer citations. Such practices perpetuate the risks and run counter to the principles of the *San Francisco Declaration on Research Assessment* (DORA; http://www.ascb.org/dora/), which emphasizes that research should be evaluated based on its intrinsic quality and specific contributions to knowledge, rather than relying on journal-based metrics like the JIF, unfairly attribute the overall impact of a journal's citations to individual articles, regardless of their actual merit. Again, new questions arise: Is it a deliberate effort or choice for journals to focus on more highly cited fields? Is this an editorial decision aimed at boosting the JIF? Or might it be that the more prominent field aligns better with the editor or current reviewers?

Whatever the cause of the serious imbalance presented in this study, researchers and scientific agencies should be aware of the existing bias. At the very least, this awareness might enable authors to make better-informed decisions about where to publish and help agencies evaluate researchers more fairly. Naturally, *Nature*, *Science*, *PNAS*, and other journals labeled as multidisciplinary are free to prioritize any selected group of areas of knowledge they wish. However, given the wide-ranging impact of articles published in these journals (e.g., see [5]), this imbalance could have far-reaching implications not only within purely scientific fields but also across political, economic, and social spheres. Fortunately, we observed encouraging signs of a general trend toward strengthening the multidisciplinary character of multidisciplinary journals over time, which could help mitigate the issues discussed here and contribute to the broader advancement of scientific knowledge.

## Supporting information

**S1 Fig. Number of publications of each research area in top multidisciplinary journals: a)** *Nature*, **b)** *Science*, **and c)** *PNAS*. Only the 15 top-ranked research areas are shown, decreasingly sorted by the median. Research areas are classified by branches of knowledge and points are colored according to year. Note that the axes are represented at different scales.
(TIF)

**S2 Fig.** Position of each research area in the ranking of the number of publications in top multidisciplinary journals; **a)** *Nature*, **b)** *Science*, and **c)** *PNAS*; per year. Number one represents the research area with more publications. Only the 15 top-ranked research areas are shown, decreasingly sorted by the median. Research areas are classified by branches of knowledge and points are colored according to year. Note that the axes are represented at different scales.
(TIF)

**S3 Fig.** Ratio between the number of publications of each research area in top multidisciplinary journals; **a)** *Nature*, **b)** *Science*, and **c)** *PNAS*; and the total number of publications of that research area in the world (journal/world ratio). Only the 15 top-ranked research areas are shown, decreasingly sorted by the median. Research areas are classified by branches of knowledge and points are colored according to year. Note that the axes are represented at different scales.
(TIF)

**S4 Fig. Yearly percentage of publications belonging to the top-30 research areas (according to number of publications; see Fig 2B) excluding the area "Multidisciplinary Sciences" in multidisciplinary journals and yearly percentage of publications belonging to the area "Biochemistry & Molecular Biology" in multidisciplinary journals, *Nature*, *Science*, and *PNAS*.**
(TIF)

**S1 Table. Number of multidisciplinary journals (according to the Web of Science) and number of publications ("articles" and "reviews") in multidisciplinary journals per decade and branch of knowledge.** The percentage of publications in each branch of knowledge in each decade is also shown.
(DOCX)

## Author Contributions

**Conceptualization:** Daniel Redondo-Gómez, Wenceslao Arroyo-Machado, Daniel Torres-Salinas, Antoni Margalida, Marcos Moleón.

**Data curation:** Daniel Redondo-Gómez, Wenceslao Arroyo-Machado, Marcos Moleón.

**Formal analysis:** Daniel Redondo-Gómez, Wenceslao Arroyo-Machado, Marcos Moleón.

**Investigation:** Daniel Redondo-Gómez, Wenceslao Arroyo-Machado, Daniel Torres-Salinas, Antoni Margalida, Marcos Moleón.

**Methodology:** Daniel Redondo-Gómez, Wenceslao Arroyo-Machado, Daniel Torres-Salinas, Antoni Margalida, Marcos Moleón.

**Resources:** Daniel Torres-Salinas.

**Software:** Daniel Redondo-Gómez.

**Supervision:** Daniel Torres-Salinas, Antoni Margalida, Marcos Moleón.

**Validation:** Daniel Redondo-Gómez, Wenceslao Arroyo-Machado, Daniel Torres-Salinas, Antoni Margalida, Marcos Moleón.

**Visualization:** Daniel Redondo-Gómez, Wenceslao Arroyo-Machado, Daniel Torres-Salinas, Antoni Margalida, Marcos Moleón.

**Writing – original draft:** Daniel Redondo-Gómez, Wenceslao Arroyo-Machado, Daniel Torres-Salinas, Antoni Margalida, Marcos Moleón.

**Writing – review & editing:** Daniel Redondo-Gómez, Wenceslao Arroyo-Machado, Daniel Torres-Salinas, Antoni Margalida, Marcos Moleón.

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
