## [Decision Letter · Decision Letter 0]

26 Feb 2024

PONE-D-23-41614A long-term assessment of the multidisciplinary degree of multidisciplinary journalsPLOS ONE

Dear Dr. Redondo-Gómez,

Thank you for submitting your manuscript to PLOS ONE. After careful consideration, we feel that it has merit but does not fully meet PLOS ONE’s publication criteria as it currently stands. Therefore, we invite you to submit a revised version of the manuscript that addresses the points raised during the review process.

We look forward to receiving your revised manuscript.

Kind regards,

Robin Haunschild

Academic Editor

PLOS ONE

Journal Requirements:

"DRG was funded by a predoctoral grant from the Junta de Andalucía (PREDOC_00262). WAM was funded by FPU Grant (FPU18/05835) from the Spanish Ministry of Universities. MM was partly supported by the Severo Ochoa Program for Centres of Excellence in R+D+I (SEV-2012-0262) and by a research contract Ramón y Cajal from the MINECO (Ministerio de Economía, Industria y Competitividad; RYC-2015-19231)."

"DRG was funded by a predoctoral grant from the Junta de Andalucía (PREDOC_00262). WAM was funded by FPU Grant (FPU18/05835) from the Spanish Ministry of Universities. MM was partly supported by the Severo Ochoa Program for Centres of Excellence in R+D+I (SEV-2012-0262) and by a research contract Ramón y Cajal from the MINECO (Ministerio de Economía, Industria y Competitividad; RYC440 2015-19231)"

"DRG was funded by a predoctoral grant from the Junta de Andalucía (PREDOC_00262). WAM was funded by FPU Grant (FPU18/05835) from the Spanish Ministry of Universities. MM was partly supported by the Severo Ochoa Program for Centres of Excellence in R+D+I (SEV-2012-0262) and by a research contract Ramón y Cajal from the MINECO (Ministerio de Economía, Industria y Competitividad; RYC-2015-19231)."

5. We notice that your supplementary figures are uploaded with the file type 'Figure'. Please amend the file type to 'Supporting Information'. Please ensure that each Supporting Information file has a legend listed in the manuscript after the references list.

Reviewers' comments:

Reviewer's Responses to Questions

**Comments to the Author**

1. Is the manuscript technically sound, and do the data support the conclusions?

Reviewer #1: Partly

Reviewer #2: Partly

2. Has the statistical analysis been performed appropriately and rigorously? 

Reviewer #1: Yes

Reviewer #2: Yes

3. Have the authors made all data underlying the findings in their manuscript fully available?

Reviewer #1: Yes

Reviewer #2: Yes

4. Is the manuscript presented in an intelligible fashion and written in standard English?

Reviewer #1: Yes

Reviewer #2: Yes

5. Review Comments to the Author

Reviewer #1: This paper examines the disciplinary affiliation of papers published in journal classified as multidisciplinary by Web of Science. This has been done before, and those papers are referenced. This paper looks at time trends and more journals.

There seem to be footnotes. At least there are several superscript numbers at the end of sentences. However, I found no footnote text in the document.

The research area of papers is analyzed here. Web of Science assigns research area to papers. How do they do this? One assumes they assign journals to research areas and then all papers in the journal are thereby assigned to the research area of the journal. But this cannot be what is happening here because all journals analyzed are classified as multidisciplinary. So the method used by Web of Science to assign papers to fields needs to be explained. Particular attention needs to be paid to what the multidisciplinary category means at the paper level.

The biggest problem with the paper is its motivation and conclusions. The motivation statement reads:

“Multidisciplinary journals, i.e., those listed in the Journal Citation Reports (JCR) under the category ‘Multidisciplinary Sciences’ or in Scopus in the subject area ‘Multidisciplinary’, are supposed to evenly support the publication of research papers from various areas of research or disciplines.”

This is not true. Nobody says any journal is “supposed to evenly support” anything. Clarivate simply cannot put these journals in any of its other categories. The journals themselves don’t care about Clarivate’s categories. The aims and scope statements of the journals would be the definitive source of expectations for each journal. Those aims and scope statements do not say they aim to evenly support all disciplines. There is no normative expectation to this effect anywhere except in the first sentence of this paper. The paper cannot be motivated by revealing that imaginary expectations are not being fulfilled.

This means that the conclusions of the paper are not supported by the analysis. The manuscript says that authors and editors should be concerned for several reasons:

1 “terminological confusion because many journals whose scientific production is highly biased towards a single area are under the label of ‘multidisciplinary’ – authors and editors are absolutely clear on what Nature, Science and PNAS stand for because each journal has a statement of its aims and scope on its website. Clarivate’s “multidisciplinary” category is of no relevance to authors and editors.

2 “some multidisciplinary articles that do not fit in specialized journals have a significant barrier to being published since they have to compete with highly specialized articles” The aims and scope of these journals say they publish across fields, as well as multidisciplinary articles. They do not say that they aim to “provide cohesion among disciplines.” That is simply the structure of the journals. An analysis of the fields of papers in those journals does not provide grounds for arguing that those journals should do something different. Any argument about what the journals should do must engage with the statements of aims and scope of the journals, which are not discussed in this paper.

3 probable DORA violation – neither DORA nor journals’ aims and scope are discussed in the paper. Therefore, the analysis provides no support for this conclusion.

4 attention devoted to different disciplines does not meet society’s demands. Society’s demands are not analyzed in this paper. Therefore the analysis provides no support for this conclusion.

That the authors erroneously believe that the “multidisciplinary” characterization has something to do with the journal editors is established definitively in the final paragraph which says: [Nature, Science and PNAS] “should consider whether the label ‘Multidisciplinary’ is a faithful reflection of the journals’ scope.” Since the journals had nothing to do with the construction of that category in Web of Science, this sentence makes no sense.

The manuscript needs to restrict itself to drawing conclusions based on its analysis.

The strength of the multidisciplinary category of papers in these journals was striking but not properly addressed in the text. The definition of this category of papers needs to be explained, and its strength in these journals needs to be fully explored and acknowledged.

If the authors were to examine the aims and scope statement of these journals, they would find that the journals aim to publish papers with broad implications across areas of research. If it hasn’t been done already, the extent to which they achieve this goal could be tested by comparing the breadth of fields citing papers in Nature that are, for example, classified into molecular biology, with the breadth of citing of papers published in the top specialist molecular biology journals. If there is no difference in breadth of scope, then Nature etc. just compete with specialist journals, and what is the point then except to foster an impact factor competition by aggregating the highest impact papers in one place, which could be argued to be detrimental compared to the alternative of having such papers scattered across more journals and so not having such and extreme gradient. That would be an interesting analysis.

Reviewer #2: The study examines the levels of multidisciplinarity of multidisciplinary journals. This is an interesting and relevant topic. However, before the article is ready for publications the authors need to address a number of outstanding questions.

1. The key to the findings is classification of articles in the journals labeled "multidisciplinary" in Web of Science. The authors should provide a more detailed description in this paper (I know they point the reader to a separate publication) so that the reader would be able to reconstruct the study. In addition, the classification they use is a bit confusing given their definition of multidisciplinarity of a journal (that it either publishes papers from number of different disciplines or it publishes multidisciplinary papers). How are then individual papers labeled as multidisciplinary? This is very important to understand given that this category of papers (not journals) is in the findings.

2. Related tot his is the definition of "areas" that the authors use. From the operational definition it looks like these are categories of journals.

3. On page 9 the description behind Figure 2 is not clear. The authors say "the 30 areas with more publications". More than what? Also, in this part on analysis focus on absolute numbers of papers is not very useful and/or informative. I would recommend using the measure the authors also introduce and that is numbers of papers per area relative to the production in each area.

4. The authors mention on page 10 the reversal of the trend regarding coverage of top-30 areas. What is the possible interpretation?

5. On page 17 the authors introduce in the discussion "interdisciplinary research." I recommend they stay with "multidisciplinary". Otherwise, they need to provide description of how the two are different in this particular study.

6. Captions for figures need to be re-visited. The captions withing the manuscript seem to be representative of what is in figures. However, captions on page 23 do not match what is on figures.

7. Figures need to be improved. For example: x- and y- axes on Figure 2a need to be labeled. Figures 2b and 2c are difficult to interpret. They should be more legible and of higher quality. x-axis should be labeled on Fig 3a and 3b. Also the x-axes should be the same for Fib 3a and 3b. In Fig 4b, c, d, e the authors need to label the axes.

6. PLOS authors have the option to publish the peer review history of their article (what does this mean?). If published, this will include your full peer review and any attached files.

Reviewer #1: No

Reviewer #2: No

---

## [Author Response · Author response to Decision Letter 0]

3 May 2024

Reviewer #1 

Summary of our responses: In response to your feedback, we made several key revisions and clarifications to our manuscript, including: elaborating on the methodology used by Web of Science for assigning research areas to papers, especially within multidisciplinary journals; clarifying our motivations, assumptions and expectations to more accurately reflect our objectives; vigorously defending our conclusions with detailed references to sections and figures that demonstrate a bias towards certain research areas in multidisciplinary journals; acknowledging the importance of both the scope and classification of journals in shaping perceptions and guiding future research towards examining the alignment of publication patterns with societal demands for research; and finally, correcting wrongly formatted references to address the issue of missing footnotes. These steps collectively enhance the clarity, accuracy, and comprehensiveness of our manuscript.

Below are our responses to your comments:

This paper examines the disciplinary affiliation of papers published in journal classified as multidisciplinary by Web of Science. This has been done before, and those papers are referenced. This paper looks at time trends and more journals.

**Our response: Thank you for the time invested in reviewing our manuscript and for your comments. In effect, our manuscript goes beyond previous studies by incorporating additional data and integrating the temporal dimension.

There seem to be footnotes. At least there are several superscript numbers at the end of sentences. However, I found no footnote text in the document.

**Our response: This was a mistake, so we apologize for it. Our ms has no footnotes; rather, several references were wrongly formatted, but they have been corrected in the new version of the manuscript.

The research area of papers is analyzed here. Web of Science assigns research area to papers. How do they do this? One assumes they assign journals to research areas and then all papers in the journal are thereby assigned to the research area of the journal. But this cannot be what is happening here because all journals analyzed are classified as multidisciplinary. So the method used by Web of Science to assign papers to fields needs to be explained. Particular attention needs to be paid to what the multidisciplinary category means at the paper level.

**Our response: Web of Science assigns a research area to each paper, regardless of the area to which the journal belongs (this is true for journals of all areas, not only “multidisciplinary” journals). We have explained this in the new version of the ms (lines 96-97).

The biggest problem with the paper is its motivation and conclusions. The motivation statement reads: “Multidisciplinary journals, i.e., those listed in the Journal Citation Reports (JCR) under the category ‘Multidisciplinary Sciences’ or in Scopus in the subject area ‘Multidisciplinary’, are supposed to evenly support the publication of research papers from various areas of research or disciplines.” This is not true. Nobody says any journal is “supposed to evenly support” anything. Clarivate simply cannot put these journals in any of its other categories. 

**Our response: We have reworded the first paragraph of the Introduction, so we hope to be more precise now (note that stating that journals “are supposed to evenly support […]” is not our motivation, but an assumption; in any case, we have removed the word “evenly”). As evident throughout the manuscript, our motivation stems from the observation that publications in journals labeled as “multidisciplinary” are strongly biased towards a limited number of research areas, specifically those with the highest citation rates. Thus, our main goals are to explore the extent to which these journals are multidisciplinary and to examine how their multidisciplinary character has changed over the long-term.

The journals themselves don’t care about Clarivate’s categories. 

**Our response: Here, we strongly disagree with you. Many journals are deeply concerned about Clarivate’s categories indeed, because they directly influence their ranking, audience, reach and, ultimately, their impact and success. In this line, see our section “How multidisciplinary are multidisciplinary journals, and its relation to the journal impact” and considerations given in the third paragraph of the Discussion.

The aims and scope statements of the journals would be the definitive source of expectations for each journal. Those aims and scope statements do not say they aim to evenly support all disciplines. There is no normative expectation to this effect anywhere except in the first sentence of this paper. The paper cannot be motivated by revealing that imaginary expectations are not being fulfilled.

**Our response: First, both the a) scope of a journal and b) its category (e.g., according to Clarivate) are sources of information that may condition the authors’ decisions on where to submit their studies. We agree that many of these “multidisciplinary” journals clearly state their directions in their scope, but in many cases the category to which the journal belongs is more used (not only by authors) than the description given in its scope. For example, many institutions and science agencies use these categories to make rankings, evaluate journals and even select personnel within the academic career. Thus, the actual scope of a journal and its assigned category should be aligned accordingly. We have now included these ideas in the Discussion (lines 403-406).

Second, one of our aims is to determine if the representation of the different branches and areas of knowledge in multidisciplinary journals is proportional to two null expectations indicating multidisciplinarity. These two expectations are not impositions of what a multidisciplinary journal should be, but starting points to better describe and understand the extent to which these journals are multidisciplinary.

This means that the conclusions of the paper are not supported by the analysis. 

**Our response: Saying that our conclusions are not supported by our results is, in our opinion, an unsubstantiated and inappropriate criticism. Our conclusions are entirely supported by our analysis and results. In short, we show that publications of multidisciplinary journals are heavily biased towards certain research areas, and that there is a general tendency in recent years to reduce this bias. All our analyses support these statements in a clear-cut way. Please see e.g. sections “Relative role of research branches and areas in the ‘Multidisciplinary Sciences’ category” and “Multidisciplinarity in top multidisciplinary journals”, as well as Figs. 1-3, in which this asymmetry between research areas is evident.

The manuscript says that authors and editors should be concerned for several reasons: 1 “terminological confusion because many journals whose scientific production is highly biased towards a single area are under the label of ‘multidisciplinary’ – authors and editors are absolutely clear on what Nature, Science and PNAS stand for because each journal has a statement of its aims and scope on its website. Clarivate’s “multidisciplinary” category is of no relevance to authors and editors.

**Our response: We totally disagree with the statement that “Clarivate’s multidisciplinary category is of no relevance to authors and editors”. As we stated above, both the a) scope of a journal and b) its category (e.g., according to Clarivate) are sources of information that may condition the authors’ decisions on where to submit their studies. We agree that many of these “multidisciplinary” journals state clear their directions in their scope, but in many cases the category to which the journal belongs is more used (not only by authors) than the description in its scope. As stated above, many science institutions and agencies utilize these categories for making rankings, journal evaluations, and even personnel selection within the academic career. Thus, the actual scope of a journal and its assigned category should be aligned accordingly.

2 “some multidisciplinary articles that do not fit in specialized journals have a significant barrier to being published since they have to compete with highly specialized articles” The aims and scope of these journals say they publish across fields, as well as multidisciplinary articles. They do not say that they aim to “provide cohesion among disciplines.” That is simply the structure of the journals. An analysis of the fields of papers in those journals does not provide grounds for arguing that those journals should do something different. Any argument about what the journals should do must engage with the statements of aims and scope of the journals, which are not discussed in this paper.

**Our response: Multidisciplinary articles are often difficult to publish in specialized journals. The only journals that may provide some publication space for these articles are “multidisciplinary” journals. However, we (and other studies before) found that articles published in “multidisciplinary” journals are highly biased towards a few highly cited areas. As we say in our manuscript, “some multidisciplinary articles that do not fit in specialized journals have a significant barrier to being published since they have to compete with highly specialized articles THAT ARE MUCH MORE ATTRACTIVE FOR THE SUCCESS OF THE JOURNAL” (lines 406-409). The part highlighted in capital letters aren’t stated in the aims and scope of these journals. Indeed, this finding is of concern not only for multidisciplinary articles, but also for articles of areas that are not preferred by these journals. Thus, this leads us to also state that “this polarization jeopardizes some of the important functions that multidisciplinary journals could fulfill, such as stimulating innovative ideas or providing cohesion among disciplines” – note that this is just a personal suggestion.

3 probable DORA violation – neither DORA nor journals’ aims and scope are discussed in the paper. Therefore, the analysis provides no support for this conclusion.

**Our response: As mentioned in previous comments, our focus is not the scope of journals but their classification. Although the scope of all journals was appropriate, the problem about their classification would still exist and, therefore, all the problems associated with it. Throughout the ms, we provide sufficient arguments to suggest that “focusing recurrently on highly cited and populated fields such as those abovementioned, which probably respond to the race towards increasing journals’ impact factor, may contravene the essence of the San Francisco Declaration on Research Assessment (DORA; http://www.ascb.org/dora/)” (lines 415-418). We think that most researchers are aware of the main points of DORA, so we don’t see the need to discuss it further. However, we are open to do it if the Editor consider it appropriate.

4 attention devoted to different disciplines does not meet society’s demands. Society’s demands are not analyzed in this paper. Therefore the analysis provides no support for this conclusion.

**Our response: We are not concluding that attention devoted to the different research branches and areas in multidisciplinary journals does not meet the demands of society, but that this issue deserves further investigation. We literally say: “whether the attention devoted to the different research branches and areas in multidisciplinary journals meets the demands of society and the planet is highly questionable and deserves in-depth analysis” (lines 418-421). Thus, this is just a suggestion for future research and does not need to be supported by any specific analysis.

That the authors erroneously believe that the “multidisciplinary” characterization has something to do with the journal editors is established definitively in the final paragraph which says: [Nature, Science and PNAS] “should consider whether the label ‘Multidisciplinary’ is a faithful reflection of the journals’ scope.” Since the journals had nothing to do with the construction of that category in Web of Science, this sentence makes no sense.

**Our response: We think that this appreciation is very appropriate, so we have reworded the sentence to “those organizations aimed at classifying and rating scientific journals should consider whether the label ‘Multidisciplinary’ accurately reflects these journals’ scope” (lines 430-432). That said, in our opinion, the “multidisciplinary” characterization has definitely something to do with the editors, as we have explained above.

The manuscript needs to restrict itself to drawing conclusions based on its analysis.

**Our response: As we explained in a previous response, we totally disagree with this statement, as our conclusions are entirely supported by our analyses and results (see above).

The strength of the multidisciplinary category of papers in these journals was striking but not properly addressed in the text. The definition of this category of papers needs to be explained, and its strength in these journals needs to be fully explored and acknowledged.

**Our response: First, the “Multidisciplinary Sciences” area is (and should be) obviously well-represented in “multidisciplinary” journals, so we think it is not interesting to analyze it in depth beyond what is necessary to understand its representation in these journals, which is indicated in the text (e.g. see lines 197-199). Indeed, what may be more striking is that only 12% of publications in these journals are within the multidisciplinary area, and that this area is not normally ranked first among all research areas represented in multidisciplinary journals. We have added some words in this regard (lines 348-350).

Second, we cannot provide a definition of this category because the classification is made by Web of Science, as it is now specified in “Material and methods” (lines 96-97). 

If the authors were to examine the aims and scope statement of these journals, they would find that the journals aim to publish papers with broad implications across areas of research. If it hasn’t been done already, the extent to which they achieve this goal could be tested by comparing the breadth of fields citing papers in Nature that are, for example, classified into molecular biology, with the breadth of citing of papers published in the top specialist molecular biology journals. If there is no difference in breadth of scope, then Nature etc. just compete with specialist journals, and what is the point then except to foster an impact factor competition by aggregating the highest impact papers in one place, which could be argued to be detrimental compared to the alternative of having such papers scattered across more journals and so not having such and extreme gradient. That would be an interesting analysis.

**Our response: We appreciate your very interesting proposal, which could be explored in future research. We believe that our current manuscript is already a robust and standalone piece of work.

Reviewer #2

Summary of our responses: In response to your feedback, we meticulously refined our manuscript by providing an improved description of the classification of publications in “multidisciplinary” journals, and clarifying the use of Web of Science classifications for both journals and publications. We also standardized our terminology, consistently opting for “multidisciplinary” to avoid confusion, improved the clarity and accuracy of figure captions, and checked the legibility and interpretative quality of our figures. All these efforts ensures the full replicability of our study.

Below are our responses to your comments:

The study examines the levels of multidisciplinarity of multidisciplinary journals. This is an interesting and relevant topic. However, before the article is ready for publications the authors need to address a number of outstanding questions.

**Our response: Thank you for the time invested in reviewing our manuscript and for your constructive comments.

1. The key to the findings is classification of articles in the journals labeled "multidisciplinary" in Web of Science. The au

---

## [Decision Letter · Decision Letter 1]

9 Jun 2024

PONE-D-23-41614R1A long-term assessment of the multidisciplinary degree of multidisciplinary journalsPLOS ONE

Dear Dr. Redondo-Gómez,

Thank you for submitting your manuscript to PLOS ONE. After careful consideration, we feel that it has merit but does not fully meet PLOS ONE’s publication criteria as it currently stands. Therefore, we invite you to submit a revised version of the manuscript that addresses the points raised during the review process.

We look forward to receiving your revised manuscript.

Kind regards,

Robin Haunschild

Academic Editor

PLOS ONE

Reviewers' comments:

Reviewer's Responses to Questions

**Comments to the Author**

1. If the authors have adequately addressed your comments raised in a previous round of review and you feel that this manuscript is now acceptable for publication, you may indicate that here to bypass the “Comments to the Author” section, enter your conflict of interest statement in the “Confidential to Editor” section, and submit your "Accept" recommendation.

Reviewer #1: (No Response)

Reviewer #2: (No Response)

2. Is the manuscript technically sound, and do the data support the conclusions?

Reviewer #1: Partly

Reviewer #2: Partly

3. Has the statistical analysis been performed appropriately and rigorously? 

Reviewer #1: Yes

Reviewer #2: Yes

4. Have the authors made all data underlying the findings in their manuscript fully available?

Reviewer #1: No

Reviewer #2: No

5. Is the manuscript presented in an intelligible fashion and written in standard English?

Reviewer #1: Yes

Reviewer #2: Yes

6. Review Comments to the Author

Reviewer #1: The manuscript has been much improved and is more moderated in its claims. However, in my view the response to my issues has been partial, and issues are still present though at a reduced level of severity.

There is still an issue of explaining how papers are classified into research areas. The current text says this: “note that the research area of the publications is assigned by the Web of Science independently of the area of the journal in which they are published”

The response to reviewers says this: “Our response: Web of Science assigns a research area to each paper, regardless of the area to which the journal belongs (this is true for journals of all areas, not only

“multidisciplinary” journals). We have explained this in the new version of the ms (lines 96-97).”

The text now clarifies that WoS classifies papers individually to areas and this is where the area classification of papers used comes from. However, the text does not explain how this classification is done. Some explanation of how this is done can be found here: https://incites.help.clarivate.com/Content/Indicators-Handbook/ih-document-reclassification.htm

In Clarivate’s explanation are several important methodological points that the reader needs to understand in order to understand what is done in this paper. These are:

1 Clarivate assigns papers “using information found in the cited references of each publication” to make an algorithmic assignment

2 “In cases where it is not possible to accurately reassign publications (e.g., when the article does not have cited references), the articles are left as multidisciplinary.” This is important because it speaks to the meaning of the “multidisciplinary” category accounting for 12% of papers in these journals. In my first set of comments I thought this category might be important. If these are largely papers without references, this category is likely of little interest. Also, the text talks about the difficulty of multidisciplinary papers finding a place to publish because in multidisciplinary journals they compete with molecular biology for space. The 12% of papers that are “multidisciplinary” seems to justify this point if we do not know what “multidisciplinary” actually means at the paper level. But if “multidisciplinary” at the paper level largely means papers with no references, then it is unclear if real multidisciplinary papers are even found in these journals at all.

3 In the response to reviewer the authors say WoS classifies all papers individually. Clarivate apparently does not do this. Instead, they say: “We apply reclassification to articles in the categories of Multidisciplinary Sciences and Medicine, General, and Internal”

Second point is to encourage more clarity in thinking about who is responsible for what. The paper seeks to go beyond description and make normative claims. That is, somebody is doing something wrong and should do better. This requires more work than is done here. Greater precision and clarity of thought would improve the paper and the conclusions drawn. Here are a few examples of what I mean.

Lines 403-406 say: “Many science institutions and agencies utilize these categories for making rankings, journal evaluations, and even personnel selection within the academic career. Thus, the actual scope of a journal and its assigned category should be aligned accordingly”

Use of passive voice in the second sentence conceals who is responsible. Presumably, Clarivate should align things. However, one could equally say that agencies and universities shouldn’t use this category. Or point out that places looking for Nature, Science or PNAS papers are essentially looking for molecular biologists and should understand that.

Similarly, I said the importance of the WoS categories to authors is overblown in the paper. In the response the authors disagree, saying authors use them to figure out where to publish. The trouble is, this paper mostly focuses on Nature, Science and PNAS. And while it is true that when I write an information science paper, for example, I may look at the WoS categories to find a journal, nobody does that and decides to submit to Nature, Science and PNAS. Instead, they have a lifelong ambition to publish in Nature, Science or PNAS and think they may have at last produced something that could be published there. So while in general categories might have some use, they are of little use to authoris in relation to the journals focused on in this paper.

On the DORA point, line 417, DORA was about research assessment, not about scope of journals. A journal seeking to publish high impact research therefore does not contravene DORA because the journal is not performing research assessment, as is required to be relevant to DORA.

Line 420, on planetary relevance, “the attention devoted to the different research branches and areas in multidisciplinary journals meets the demands of society and the planet is highly questionable. . .” I suggested this statement was not supported by the analyses. The response points to the final words about doing more research to say it is fine. I re-emphasis here the judgement – “questionable” – and that nothing in the analysis presented supports that point. Of course, life, the universe and everything deserves further research, but that still doesn’t justify making unsupported claims.

Reviewer #2: My major questions/concerns are still not addressed. Their response to my request to provide more details so that the reader could reconstruct the study: "our methodological procedures are totally replicable by any other researchers...The other classifications come from the Web of Science, the Journal Citation Reports, and the InCites Dataset. As these databases are well-known by scientific readership, we believe that it is not necessary to explain details about their functioning" is not satisfactory. Despite having extensive experience in this area I would not be able to reproduce the study. Also given that the main topic of their paper deals with classification they need to explain the classification(s) used by Clarivate.

First, the authors should specify after they obtained ISSNs using the Journal Citation Reports what database they used (Web of Science or InCites Dataset) to collect data on publications. They should then specify what fields they used. My best guess is that they used: PY (Year Published), SO (Publication Titles), SU (Research Area) and, WC (Web of Science Categories) fields. However, they should specify this. They should also then say something about Web of Science categories - how were they constructed, how many are there, how they address the scenario of multiple subject categories being assigned to a journal. They also say in the methods that they included in the data set all the journals that have at least one of their editions in the period between 1997 and 2020 classified as "Multidisciplinary Sciences." What are the implications? How many such journals are there? Is there a difference between journals being classified in this category throughout the period of the study and the ones that were classified only occasionally? Also, are there journals that have multiple Web of Science categories assigned to them? If so, how many? The authors should also provide more details about the field SU (Research Area). How many Research Areas are there in Web of Science? How are they assigned? Does a paper have multiple Research Areas? How is such a case resolved in this study?

To avoid confusion maybe the authors should use different terminology when they talk about classes for journals and papers.

Finally, the authors should provide tables that list "branches of knowledge" and Web of Science categories assigned to each.

These clarifications are essential before the findings of the study can be properly assessed.

A few additional comments:

In lines 52-53 the authors say: "multidisciplinary journals usually tend to accept articles based on their potential citability (4,5)". I don't think that these references provide claims that the criterion for the acceptance of papers by journals is the potential citability of articles. This is a very strong claim which is not founded in the supported evidence.

In lines 63-64 the authors say: "the growing competitiveness of the increasingly crowded multidisciplinary discipline may jeopardize its multidisciplinary nature." What is "multidisciplinary discipline"?

In line 82 the authors say: "the proportion of publications in the journals that belong to the top area". How do they define top area (the one with the largest number of articles or the one with most citations)?

In lines 226-232 the authors say "Number one represents the research area with more publications." What is "number one" what is "the research area with more publications"? Also why the x axis of Figure 2b goes from 1 to 8000 and Figure 2c goes from 150 to 1. I know understand that the scales are different, but the axes should be consistent (either from smallest to the largest or from the largest to the smallest number). As it comes to Figure 2a I am not sure what the major take-away message is, nor what do bars represent.

The authors have still not labeled x-axes for Figure 3a and 3b, so I don't know what number 0, 5, 10 and 0, 1, 2, 3, 4, 5, 6, 7, 8 represent. There is no reason for Figure 3a to have different units than the one on 3b. Also as far as I can see in both 3a and 3b there is no area that has value above 5, so why does one go to 10 and another to 8? Both should use 0, 1, 2, 3, 4, 5, 6.

Finally, the authors claim that: "All data are available from the InCites dataset (https://incites.clarivate.com/)" and the data is fully available without restriction. However, when I followed the link I was asked for the subscription.

7. PLOS authors have the option to publish the peer review history of their article (what does this mean?). If published, this will include your full peer review and any attached files.

Reviewer #1: No

Reviewer #2: No

---

## [Author Response · Author response to Decision Letter 1]

8 Aug 2024

Reviewer #1: The manuscript has been much improved and is more moderated in its claims. However, in my view the response to my issues has been partial, and issues are still present though at a reduced level of severity.

**Our response: Thank you for the time invested in this new round of revision. We hope the new version has fully resolved the issues raised.

There is still an issue of explaining how papers are classified into research areas. 

**Our response: To better contextualize and inform readers about how papers are classified into research areas, we have included a new subsection in the methodology. This subsection explains in detail how Web of Science and InCites classify journals and articles by Web of Science categories, and how the latter reassigns publications to other categories (lines 85-100). Additionally, we have improved the description of the data processing procedure based on InCites (lines 115-118).

The current text says this: “note that the research area of the publications is assigned by the Web of Science independently of the area of the journal in which they are published”

**Our response: We have rewritten the sentence to clarify any doubts and provide more details about the classification system (lines 116-118).

The response to reviewers says this: “Our response: Web of Science assigns a research area to each paper, regardless of the area to which the journal belongs (this is true for journals of all areas, not only “multidisciplinary” journals). We have explained this in the new version of the ms (lines 96-97).” The text now clarifies that WoS classifies papers individually to areas and this is where the area classification of papers used comes from. However, the text does not explain how this classification is done. Some explanation of how this is done can be found here: https://incites.help.clarivate.com/Content/Indicators-Handbook/ih-document-reclassification.htm

**Our response: Thank you for the suggestion. As mentioned above, we have included a new subsection with all those details (“Web of Science categories”) in the methodology (lines 85-100). Moreover, we have specified in the “Data gathering” section that the WoS algorithmically make the classification of publications for multidisciplinary journals according to cited references, when available (lines 115-118).

In Clarivate’s explanation are several important methodological points that the reader needs to understand in order to understand what is done in this paper. These are:

1 Clarivate assigns papers “using information found in the cited references of each publication” to make an algorithmic assignment.

**Our response: We have included a new subsection with that information (“Web of Science categories”) in the methodology (lines 85-100). Moreover, we have mentioned the algorithmic reassignment in the “Data gathering” section (lines 115-118). In any case, regardless of the internal classification system, what we are addressing here is the outcome of the classification and its effects on different scales (see lines 421-443).

2 “In cases where it is not possible to accurately reassign publications (e.g., when the article does not have cited references), the articles are left as multidisciplinary.” This is important because it speaks to the meaning of the “multidisciplinary” category accounting for 12% of papers in these journals. In my first set of comments I thought this category might be important. If these are largely papers without references, this category is likely of little interest. 

**Our response: We have specified in the new subsection (lines 85-100) and in the “Data gathering” section (lines 115-118) that the reassignment is only made for some journals and when references are available. We are aware of this issue, but it is of little significance (not many publications do not have references) and does not affect our results and conclusions. Moreover, our approach is conservative, and the potential bias you mentioned only reinforces our message. If these unclassified papers were reclassified into other categories, the imbalances we are illustrating would likely be even more pronounced.

Also, the text talks about the difficulty of multidisciplinary papers finding a place to publish because in multidisciplinary journals they compete with molecular biology for space. The 12% of papers that are “multidisciplinary” seems to justify this point if we do not know what “multidisciplinary” actually means at the paper level. But if “multidisciplinary” at the paper level largely means papers with no references, then it is unclear if real multidisciplinary papers are even found in these journals at all.

**Our response: We understand your reasoning. However, our statement about the difficulty of multidisciplinary papers finding a place to publish is not based on that percentage, but on the strong disbalance to highly cited categories. It is evident that if most multidisciplinary journals publish most of their articles in this “Biochemistry and Molecular Biology”, the space left for other types of articles, including multidisciplinary ones, is very limited.

3 In the response to reviewer the authors say WoS classifies all papers individually. Clarivate apparently does not do this. Instead, they say: “We apply reclassification to articles in the categories of Multidisciplinary Sciences and Medicine, General, and Internal”

**Our response: We have specified in the new subsection (lines 85-100) and in the “Data gathering” section (lines 115-118) that the WoS algorithmically make the classification of publications for multidisciplinary journals according to cited references, when available.

Second point is to encourage more clarity in thinking about who is responsible for what. The paper seeks to go beyond description and make normative claims. That is, somebody is doing something wrong and should do better. This requires more work than is done here. Greater precision and clarity of thought would improve the paper and the conclusions drawn. Here are a few examples of what I mean. Lines 403-406 say: “Many science institutions and agencies utilize these categories for making rankings, journal evaluations, and even personnel selection within the academic career. Thus, the actual scope of a journal and its assigned category should be aligned accordingly” Use of passive voice in the second sentence conceals who is responsible. Presumably, Clarivate should align things. However, one could equally say that agencies and universities shouldn’t use this category. Or point out that places looking for Nature, Science or PNAS papers are essentially looking for molecular biologists and should understand that.

**Our response: Here, we would like to remark that “thinking about who is responsible for what” is not our mission. We just aim to highlight that something is wrong and should be considered by the scientific community. Whether the effort in aligning the actual scope of a journal and its assigned category should be made by Clarivate, the agencies, and/or the journals is a matter of further discussion that goes beyond our study. We have added this sentence to be clear (lines 427-428).

Similarly, I said the importance of the WoS categories to authors is overblown in the paper. In the response the authors disagree, saying authors use them to figure out where to publish. The trouble is, this paper mostly focuses on Nature, Science and PNAS. And while it is true that when I write an information science paper, for example, I may look at the WoS categories to find a journal, nobody does that and decides to submit to Nature, Science and PNAS. Instead, they have a lifelong ambition to publish in Nature, Science or PNAS and think they may have at last produced something that could be published there. So while in general categories might have some use, they are of little use to authoris in relation to the journals focused on in this paper.

**Our response: First, we do not believe that we are exaggerating the importance of WoS categories for authors (see line 421 for the only mention of authors in the Discussion). Second, in many scientific disciplines, such as the exact sciences, WoS categories can indeed be relevant for authors, who are continuously evaluated (for contracts, projects, etc.) based on metrics such as the number/proportion of papers published in the first quartile of their categories. Third, please note that, while we discuss the cases of Nature, Science, and PNAS in detail, the article does not focus on them exclusively. This is actually one of the strengths of our study. We simply found it appropriate to dedicate a specific section to them, as they are unique and significant within this ecosystem, as you rightly point out.

On the DORA point, line 417, DORA was about research assessment, not about scope of journals. A journal seeking to publish high impact research therefore does not contravene DORA because the journal is not performing research assessment, as is required to be relevant to DORA.

**Our response: Certainly, the issue does not lie with the journal itself. We have clarified this point in the text (lines 437-443).

Line 420, on planetary relevance, “the attention devoted to the different research branches and areas in multidisciplinary journals meets the demands of society and the planet is highly questionable. . .” I suggested this statement was not supported by the analyses. The response points to the final words about doing more research to say it is fine. I re-emphasis here the judgement – “questionable” – and that nothing in the analysis presented supports that point. Of course, life, the universe and everything deserves further research, but that still doesn’t justify making unsupported claims.

**Our response: Given that this sentence appears to cause confusion and does not contribute any significant ideas to the paper, we have removed it. 

Reviewer #2: My major questions/concerns are still not addressed. Their response to my request to provide more details so that the reader could reconstruct the study: "our methodological procedures are totally replicable by any other researchers...The other classifications come from the Web of Science, the Journal Citation Reports, and the InCites Dataset. As these databases are well-known by scientific readership, we believe that it is not necessary to explain details about their functioning" is not satisfactory. Despite having extensive experience in this area I would not be able to reproduce the study. Also given that the main topic of their paper deals with classification they need to explain the classification(s) used by Clarivate.

**Our response: Thank you once more for the time invested in reviewing our manuscript. This issue is also shared by Reviewer 1; following the suggestions of both reviewers, we have introduced an entire subsection to better describe all these aspects concerning the Web of Science categories (lines 85-100). Moreover, we have provided more details about the classification system in the “Data gathering” section (lines 115-118). In any case, please note that a better understanding of how WoS and InCites work does not affect reproducibility of our study, as any researcher who follows the procedures described in the “Data gathering” and “Data treatment and analyses” sections should obtain the same dataset and results as we did.

First, the authors should specify after they obtained ISSNs using the Journal Citation Reports what database they used (Web of Science or InCites Dataset) to collect data on publications. They should then specify what fields they used. My best guess is that they used: PY (Year Published), SO (Publication Titles), SU (Research Area) and, WC (Web of Science Categories) fields. However, they should specify this. 

**Our response: We have clarified issues related to data extraction, indicating that the data are extracted from InCites and not from Web of Science (lines 111-115).

They should also then say something about Web of Science categories - how were they constructed, how many are there, how they address the scenario of multiple subject categories being assigned to a journal. 

**Our response: As mentioned above, we have specified in the new subsection (lines 85-100) and in the “Data gathering” section (lines 115-118) all those details.

They also say in the methods that they included in the data set all the journals that have at least one of their editions in the period between 1997 and 2020 classified as "Multidisciplinary Sciences." What are the implications? How many such journals are there? Is there a difference between journals being classified in this category throughout the period of the study and the ones that were classified only occasionally? Also, are there journals that have multiple Web of Science categories assigned to them? If so, how many? 

**Our response: Of the 127 journals considered in our study, 55 (43%) were included in the category “Multidisciplinary Sciences” for at least half of the study period, and only 10 (8%) were included for <5 years. Thus, very few journals were classified occasionally as multidisciplinary. Also, during the study period, 86 journals (68%) were exclusively categorized as “Multidisciplinary Sciences”. Therefore, we can say that our selection of journals has greatly contributed to multidisciplinary sciences during our study period. For clarity, we have included these data in the new version of the ms (lines 107-111).

The authors should also provide more details about the field SU (Research Area). How many Research Areas are there in Web of Science? 

**Our response: There are 254 research areas, as we already specified in the previous versions of our ms (line 89 in the current version). 

How are they assigned? 

**Our response: Articles inherit the categories of the journal in which they are published, with some exceptions, as we now specify in the new subsection on “Web of Science categories” (lines 85-100). 

Does a paper have multiple Research Areas? How is such a case resolved in this study?

**Our response: Please note that this was already specified in the previous version of our ms (lines 135-139 in the current version): “Publications included in more than one research area were considered in each of the areas they were included (but note that only 3.1% of publications were included in more than one area, and that these multiple areas were often closely related, e.g., “Computer Science, Hardware & Architecture” and “Computer Science, Theory & Methods”).”

To avoid confusion maybe the authors should use different terminology when they talk about classes for journals and papers.

**Our response: To avoid confusion, in the new version we clarify this distinction by using "category" for the areas of the journals and "research areas" for the areas of the publications.

Finally, the authors should provide tables that list "branches of knowledge" and Web of Science categories assigned to each. 

**Our response: As specified in the text (lines 119-122), we base this classification on a previously published article where this table can be found (Arroyo-Machado & Torres-Salinas, 2021). Therefore, we do not believe it is necessary to include the same table in our article.

These clarifications are essential before the findings of the study can be properly assessed.

**Our response: We hope that our responses and the improvements made in the new version have fully resolved the issues raised.

A few additional comments:

In lines 52-53 the authors say: "multidisciplinary journals usually tend to accept articles based on their potential citability (4,5)". I don't think that these references provide claims that the criterion for the acceptance of papers by journals is the potential citability of articles. This is a very strong claim which is not founded in the supported evidence.

**Our response: In this case, we disagree with you. The references provided do fully support the message we convey. Reference 4 states: “Existing multidisciplinary journals publish selectively most-cited papers from fields with high citation density”; and reference 5 states: “Articles in them [multidisciplinary journals] are explicitly chosen not only because they move their specific literatures in important 

---

## [Decision Letter · Decision Letter 2]

7 Oct 2024

PONE-D-23-41614R2A long-term assessment of the multidisciplinary degree of multidisciplinary journalsPLOS ONE

Dear Dr. Redondo-Gómez,

Thank you for submitting your manuscript to PLOS ONE. After careful consideration, we feel that it has merit but does not fully meet PLOS ONE’s publication criteria as it currently stands. Therefore, we invite you to submit a revised version of the manuscript that addresses the points raised during the review process.

We look forward to receiving your revised manuscript.

Kind regards,

Robin Haunschild

Academic Editor

PLOS ONE

Reviewers' comments:

Reviewer's Responses to Questions

**Comments to the Author**

1. If the authors have adequately addressed your comments raised in a previous round of review and you feel that this manuscript is now acceptable for publication, you may indicate that here to bypass the “Comments to the Author” section, enter your conflict of interest statement in the “Confidential to Editor” section, and submit your "Accept" recommendation.

Reviewer #1: (No Response)

Reviewer #2: All comments have been addressed

Reviewer #3: (No Response)

2. Is the manuscript technically sound, and do the data support the conclusions?

Reviewer #1: No

Reviewer #2: Yes

Reviewer #3: Yes

3. Has the statistical analysis been performed appropriately and rigorously? 

Reviewer #1: Yes

Reviewer #2: Yes

Reviewer #3: Yes

4. Have the authors made all data underlying the findings in their manuscript fully available?

Reviewer #1: Yes

Reviewer #2: Yes

Reviewer #3: Yes

5. Is the manuscript presented in an intelligible fashion and written in standard English?

Reviewer #1: Yes

Reviewer #2: Yes

Reviewer #3: Yes

6. Review Comments to the Author

Reviewer #1: The analysis shows that life sciences is over represented in these journals compared to all of science. This is well known. The analysis also shows that this may be decreasing, which may be less well known. The methods are now better explained and the conclusions somewhat moderated.

This paper wants to conclude that Science, Nature and PNAS shouldn’t be classified as multidisciplinary because 13% of their papers are in biochemistry & molecular biology, and that is the largest WoS category in these journals. Even at the disciplinary level shown in figure 1, more than life sciences is published in these journals. How are they supposed to be classified? Given the data in the paper, it is easy to conclude that multidisciplinary is indeed an accurate classification of these journals.

Therefore, I disagree with the conclusions of the paper. There is no problem here, not for the journals, not for Clarivate, not for scientists nor for the institutions that employ them.

Reviewer #2: (No Response)

Reviewer #3: The manuscript presents an interesting study of the multidisciplinarity degree of multidisciplinary journals and changes in that degree over time. I see that the first round of reviewers' comments contain some concerns and some divided opinions, hence I've tried to give this a careful consideration.

I believe the methodology used has been appropriate and the results support the conclusions, but perhaps some of the conclusions, or the concerns described from line 400 onward, need somewhat more explanations and discussion.

I agree with the first concern, that these categories are used for many reasons and thus it would be important that they align with the journals accordingly. A mismatch between journals and their categories would lead to inaccurate rankings and evaluations. But is this a problem that the journals should try to solve or Clarivate who has assigned the categories? What level of multidisciplinarity is needed for a journal to be appropriately called multidisciplinary? And on the other hand, what level of specialization is needed for a journal to move from multidisciplinary category to a subject specific category? How many articles in Biochemistry and molecular science should Nature have for it to belong to that category? This leads to a more fundamental question of the reasons behind the degree of multidisciplinarity and connects to the second concern that the authors have stated. How much of the shown specialization of multidisciplinary journals is an intentional editorial choice by a journal? Is it the journal, editor, or reviewers that have influenced the direction the journals take? Does the degree of multidisciplinarity change when the editor changes? Is it an intentional decision that certain articles will not be published, making it difficult for more multidisciplinary articles to be published? This goes to the reasons for the changes in the degree of multidisciplinarity and I understand that this was beyond the scope of this study, but it would be interesting to read how the authors see these issues o concerns.

The third concern raised by the authors (line 415 onward) has two parts that warrant some more thought. First, is it an intentional effort, a choice, that the journals focus on more highly cited fields? Is it an editorial decision to boost the IF? Or could it be that the more prominent field aligns better with the editor or current reviewers? Again, there are probably no clear answers. But what I do think needs a bit more explanation, although most readers probably have some idea about what DORA is, is why this would be against the essence of DORA? I'd rather see this written out than leave it for the reader to contemplate. Besides, I'd very much like to read about the authors thinking on this matter. And finally the fourth concern, whether the journals meet the demands of the society and the planet. I believe this is too vague and too ideological of a statement. While this might be true, this is not something that I see evidence of in the results (as mentioning the society and the planet takes the argument to an ideological level, rather than factual).

Regarding the categorization and the journals choice to publish what they want (that fits their scope and mission statements), I fully agree that journals may shift their focus how they please and it would be the job of Clarivate and other indexes to reassign the journals if and when appropriate. But I'd like to add that readers that are not fully knowledgeable about how journals are assigned to categories, most likely expect multidisciplinary journals to be truly multidisciplinary, ie not be biased to published more articles from specific fields. Even for this reason, I think this study makes some valuable contributions.

My final point is about the choice of wording at some places. The authors write for instance "PNAS have consistently increased its degree of multidisciplinarity" (line 396). This goes back to my earlier point, whether it is a intentional effort to increase this degree or some unintentional effect of for instance the background of the editor or the reviewers. The wording that the authors have used suggests that this would be intentional, but I'm not sure if that can be read from the results. On the other hand, "the degree of multidisciplinarity in PNAS has consistently increased" would not suggest anything, it would only state that this is the case.

7. PLOS authors have the option to publish the peer review history of their article (what does this mean?). If published, this will include your full peer review and any attached files.

Reviewer #1: No

Reviewer #2: No

Reviewer #3: No

---

## [Author Response · Author response to Decision Letter 2]

31 Oct 2024

Please note that the line numbers referenced correspond to the updated version of the manuscript with track changes.

Reviewer #1: 

The analysis shows that life sciences is over represented in these journals compared to all of science. This is well known. The analysis also shows that this may be decreasing, which may be less well known. The methods are now better explained and the conclusions somewhat moderated.

**Our response: Thank you for the time dedicated to this new round of revision. We appreciate your acknowledgment of one of the values of the study, namely the temporal component, and the improvements made to clarify the methods and moderate the discussion. As you will see below, we have made an additional effort to accommodate the Discussion to your criticisms.

This paper wants to conclude that Science, Nature and PNAS shouldn’t be classified as multidisciplinary because 13% of their papers are in biochemistry & molecular biology, and that is the largest WoS category in these journals. Even at the disciplinary level shown in figure 1, more than life sciences is published in these journals. How are they supposed to be classified? Given the data in the paper, it is easy to conclude that multidisciplinary is indeed an accurate classification of these journals. Therefore, I disagree with the conclusions of the paper. There is no problem here, not for the journals, not for Clarivate, not for scientists nor for the institutions that employ them.

**Our response: We appreciate your emphasis on these comments, as they prompted us to reconsider our conclusions. Consequently, we have removed the part suggesting that organizations responsible for classifying and rating scientific journals should assess whether the label ‘Multidisciplinary’ accurately represents the scope of Nature, Science, PNAS, and other journals categorized as multidisciplinary. Instead, in the revised version, we now state that “the multidisciplinary label need not be removed from these journals” (lines 434-435), and we shift the focus to researchers and scientific agencies, encouraging them to be aware of the existing polarization of multidisciplinary journals toward specific areas (lines 477-479; we have also included this idea at the end of the Abstract, lines 40-42). Additionally, we have briefly expanded the discussion on issues arising from this bias to stimulate further debate and research on this topic (lines 427-475). Thus, we hope this revised version of the Discussion contributes to promoting more balanced publication opportunities and fairer research evaluations, rather than critiquing the editorial decisions of these journals or Clarivate’s classification efforts.

Reviewer #3: 

The manuscript presents an interesting study of the multidisciplinarity degree of multidisciplinary journals and changes in that degree over time. I see that the first round of reviewers' comments contain some concerns and some divided opinions, hence I've tried to give this a careful consideration.

**Our response: Thank you for your time and consideration. We hope that the changes made in the revised version satisfactorily address your concerns.

I believe the methodology used has been appropriate and the results support the conclusions, but perhaps some of the conclusions, or the concerns described from line 400 onward, need somewhat more explanations and discussion.

**Our response: We appreciate your recognition that the methods are appropriate and the results support the main conclusions. We have made some changes to the final section of the Discussion (especially, lines 427-498) that we hope have clarified your concerns (please see below for details).

I agree with the first concern, that these categories are used for many reasons and thus it would be important that they align with the journals accordingly. A mismatch between journals and their categories would lead to inaccurate rankings and evaluations. But is this a problem that the journals should try to solve or Clarivate who has assigned the categories? What level of multidisciplinarity is needed for a journal to be appropriately called multidisciplinary? And on the other hand, what level of specialization is needed for a journal to move from multidisciplinary category to a subject specific category? How many articles in Biochemistry and molecular science should Nature have for it to belong to that category?

This leads to a more fundamental question of the reasons behind the degree of multidisciplinarity and connects to the second concern that the authors have stated. How much of the shown specialization of multidisciplinary journals is an intentional editorial choice by a journal? Is it the journal, editor, or reviewers that have influenced the direction the journals take? Does the degree of multidisciplinarity change when the editor changes? Is it an intentional decision that certain articles will not be published, making it difficult for more multidisciplinary articles to be published? This goes to the reasons for the changes in the degree of multidisciplinarity and I understand that this was beyond the scope of this study, but it would be interesting to read how the authors see these issues o concerns.

The third concern raised by the authors (line 415 onward) has two parts that warrant some more thought. First, is it an intentional effort, a choice, that the journals focus on more highly cited fields? Is it an editorial decision to boost the IF? Or could it be that the more prominent field aligns better with the editor or current reviewers? Again, there are probably no clear answers. 

**Our response: The questions you propose are indeed thought-provoking, so we have included several of them as open questions in the new version of the Discussion to encourage debate and inspire future research on this topic (lines 451-457, 472-475). As you rightly note, there may not be definitive answers here, or at least our data do not allow us to fully resolve these questions. It does seem reasonable to suggest that this polarization may serve as a strategy to increase the impact factor. However, while this aligns with our findings, we prefer not to speculate further, leaving the door open for future debate and research (lines 451-457).

But what I do think needs a bit more explanation, although most readers probably have some idea about what DORA is, is why this would be against the essence of DORA? I'd rather see this written out than leave it for the reader to contemplate. Besides, I'd very much like to read about the authors thinking on this matter. 

**Our response: Following your suggestion, we have revised this paragraph (lines 459-475). We hope the result has addressed the issues raised.

And finally the fourth concern, whether the journals meet the demands of the society and the planet. I believe this is too vague and too ideological of a statement. While this might be true, this is not something that I see evidence of in the results (as mentioning the society and the planet takes the argument to an ideological level, rather than factual).

**Our response: We agree with you, so we have removed that sentence.

Regarding the categorization and the journals choice to publish what they want (that fits their scope and mission statements), I fully agree that journals may shift their focus how they please and it would be the job of Clarivate and other indexes to reassign the journals if and when appropriate. But I'd like to add that readers that are not fully knowledgeable about how journals are assigned to categories, most likely expect multidisciplinary journals to be truly multidisciplinary, ie not be biased to published more articles from specific fields. Even for this reason, I think this study makes some valuable contributions.

**Our response: We fully agree with your reflections. We have considered them in the new sentences at the end of the Abstract (lines 40-42) and in the final paragraph of the Discussion (lines 477-479).

My final point is about the choice of wording at some places. The authors write for instance "PNAS have consistently increased its degree of multidisciplinarity" (line 396). This goes back to my earlier point, whether it is a intentional effort to increase this degree or some unintentional effect of for instance the background of the editor or the reviewers. The wording that the authors have used suggests that this would be intentional, but I'm not sure if that can be read from the results. On the other hand, "the degree of multidisciplinarity in PNAS has consistently increased" would not suggest anything, it would only state that this is the case.

**Our response: We appreciate the suggestion. The proposed sentence conveys exactly the meaning we intend, so we have changed our original sentence accordingly (lines 423-424). Moreover, we have revised several parts of the manuscript, with special emphasis on the Abstract and the first paragraph of the Discussion, to enhance clarity.

---

## [Decision Letter · Decision Letter 3]

14 Nov 2024

A long-term assessment of the multidisciplinary degree of multidisciplinary journals

PONE-D-23-41614R3

Dear Dr. Redondo-Gómez,

We’re pleased to inform you that your manuscript has been judged scientifically suitable for publication and will be formally accepted for publication once it meets all outstanding technical requirements.

Kind regards,

Robin Haunschild

Academic Editor

PLOS ONE

Additional Editor Comments (optional):

Reviewers' comments:

Reviewer's Responses to Questions

**Comments to the Author**

1. If the authors have adequately addressed your comments raised in a previous round of review and you feel that this manuscript is now acceptable for publication, you may indicate that here to bypass the “Comments to the Author” section, enter your conflict of interest statement in the “Confidential to Editor” section, and submit your "Accept" recommendation.

Reviewer #3: All comments have been addressed

2. Is the manuscript technically sound, and do the data support the conclusions?

Reviewer #3: Yes

3. Has the statistical analysis been performed appropriately and rigorously? 

Reviewer #3: Yes

4. Have the authors made all data underlying the findings in their manuscript fully available?

Reviewer #3: Yes

5. Is the manuscript presented in an intelligible fashion and written in standard English?

Reviewer #3: Yes

6. Review Comments to the Author

Reviewer #3: The authors have addressed all concerns and comments to my satisfaction. The manuscript can be accepted for publication.

7. PLOS authors have the option to publish the peer review history of their article (what does this mean?). If published, this will include your full peer review and any attached files.

Reviewer #3: No

---

## [Editor Report · Acceptance letter]

18 Nov 2024

PONE-D-23-41614R3 

PLOS ONE

Dear Dr. Redondo-Gómez, 

I'm pleased to inform you that your manuscript has been deemed suitable for publication in PLOS ONE. Congratulations! Your manuscript is now being handed over to our production team.

Kind regards, 

on behalf of

Dr. Robin Haunschild 

Academic Editor

PLOS ONE